# Using *Bacillus subtilis* as a Host Cell to Express an Antimicrobial Peptide from the Marine Chordate *Ciona intestinalis*

**DOI:** 10.3390/md19020111

**Published:** 2021-02-12

**Authors:** Bing-Chang Lee, Jui-Che Tsai, Cheng-Yung Lin, Chun-Wei Hung, Jin-Chuan Sheu, Huai-Jen Tsai

**Affiliations:** 1Institute of Biomedical Sciences, Mackay Medical College, New Taipei City 25245, Taiwan; bingchanglee@mmc.edu.tw (B.-C.L.); cylin@mmc.edu.tw (C.-Y.L.); 2Institute of Molecular and Cellular Biology, National Taiwan University, Taipei 10617, Taiwan; r05b43023@ntu.edu.tw (J.-C.T.); r05b43033@ntu.edu.tw (C.-W.H.); 3Liver Disease Prevention and Treatment Research Foundation, Taipei 10047, Taiwan; jcsheu@ntu.edu.tw; 4Department of Life Science, Fu Jen Catholic University, New Taipei 20206, Taiwan

**Keywords:** CiMAM, antimicrobial peptide, lactoferricin, transgenic line, bactericidal activity

## Abstract

*Ciona* molecule against microbes-A24 (CiMAM) isolated from the marine chordate *Ciona intestinalis* is an antimicrobial peptide. To generate CiMAM-expressing transgenic *Bacillus subtilis*, we constructed a plasmid expressing recombinant CiMAM (rCiMAM) and introduced it into *B. subtilis.* Transgenic strains C117 and C166 were selected since they were able to highly and stably express rCiMAM. We studied the bactericidal activity of pepsin-digested extracts from rCiMAM-expressing strains against freshwater and euryhaline pathogens that commonly occur in aquaculture ponds and found no difference from that of lactoferricin-expressing strains. The bactericidal activity of 1-μL aliquot from a total 5.5 mL extracted from 5 mL of cultured C117 (1.45 × 10^8^ CFU·mL^−1^) and C166 (2.17 × 10^8^ CFU·mL^−1^) against halophilic bacteria was equivalent to the efficacy of 57.06 and 32.35 ng of Tetracycline against *Vibrio natriegens*, 47.07 and 25.2 ng against *V. parahaemolyticus*, and 58.17 and 36.55 ng against *V. alginolyticus,* respectively, indicating higher bactericidal activity of pepsin-extracts from rCiMAM-containing strains against halophilic bacteria compared to that from lactoferricin-containing strains. Since the antibacterial activity of rCiMAM-expressing *B. subtilis* strains shows higher competence against halophilic pathogens compared to that against freshwater and euryhaline pathogens, these strains are promising candidates to protect marine fish and shellfish from halophilic bacterial infection.

## 1. Introduction

The aquaculture industry is an important source of animal protein for human consumption, and among cultured aquatic species, marine organisms contribute more than half the total economic value. To meet this demand, the aquaculture industry usually employs a high-density culturing system that is known to cause physiological stress on cultured fish and shellfish. As a result, the immunity of cultured organisms is reduced, making them more vulnerable to bacterial infection, particularly during periods of rapid climatic and environmental change. Although antibiotics are generally administered to solve this problem [1], the chance of emerging antibiotic-resistant bacteria increases [2,3]. This calls for the development of a viable antimicrobial peptide (AMP) as an alternative to antibiotics applied in both animal husbandry and aquaculture farming.

*Ciona intestinalis* (Linnaeus, 1767) [4] belongs to the Phylum *Chordata*, Class *Ascidiacea*, Order *Cionida*e and Genus *Ciona*. It is a primitive marine vertebrate with a soft and transparent long oval tubular colloidal tunicate, enabling fixation of the base on seabed rocks, shells or boat hull as sessile life. Since the 19th century, *C. intestinalis* has been used as a model organism for the study of developmental and evolutionary biology [5,6]. With the development of genetics and molecular biology, Fedders et al. [7] employed the bioinformatics approach to search for potential AMPs from the expressed sequence tag (EST) database of oceanic *C. intestinalis*. A short oligopeptide with an amphipathic **α**-helix structure composed of 24 amino acid residues was obtained and named as *Ciona* molecule against microbes-A24 (CiMAM). CiMAM possesses a broad spectrum of antimicrobial activity against a variety of Gram-positive and Gram-negative bacteria. For example, Fedders et al. [8] demonstrated the bactericidal activity of CiMAM against a variety of multidrug-resistant bacteria and anaerobic pathogens. CiMAM is also shown to have considerable bactericidal effect against mycobacteria, which can hide and proliferate in macrophages [9]. Additionally, CiMAM has bactericidal ability against *Legionella pneumophila* and its natural host *Acanthamoeba castellanii* [10]. 

*Bacillus subtilis,* a Gram-positive bacterium, usually exists in the soil, the gastrointestinal tract of herbivores, and even in the gastrointestinal tract of humans [11]. *B. subtilis* has antibacterial ability since it can produce more than 24 kinds of antibacterial molecules [12]. In agriculture, *B. subtilis* is used to treat plant diseases [13,14]. Furthermore, *B. subtilis* has been considered as a microorganism generally recognized as safe (GRAS). Therefore, several commercial products use it as a probiotic for humans or animals [15]. Taking advantage of these merits, *B. subtilis* has proved to be an excellent host cell to express exogenous AMPs and can be directly added into fish meal as a feed supplement without undergoing a complicated purification process. Recently, Lee et al. [16] demonstrated that the lactoferricin-producing transgenic *B. subtilis* WB800 strain can kill two common aquatic animal pathogens, namely *E. tarda* in freshwater and *V. parahaemolyticus* in seawater. However, the lactoferricin-producing transgenic line generated by Lee et al. [16] suffers a limitation since it has less effective bactericidal activity against such halophilic bacteria as *Vibrio natrigens*. Therefore, in this study, we used CiMAM as an alternative AMP with antimicrobial activity in a high salinity environment, thus preventing marine pathogenic bacteria from infecting marine cultured fish and shellfish. 

## 2. Results

### 2.1. Construction of Expression Vector

To express exogenous recombinant CiMAM in *Bacillus subtilis,* we constructed an expression plasmid pP43-2CiMAM-GFP. After PCR amplification, we sequenced and confirmed that the PCR product obtained from 5’-end set primers was 103-bp DNA, that from the 3’-end set primers was 108-bp, while that from the last time of PCR was 183-bp (Figure 1A). The final PCR product contained two copies of CiMAM (2CiMAM) cDNAs and a codon for phenylalanine (F), which is a pepsin-specific cleavage site, inserted between 2CiMAM cDNAs, termed as the CiMAM-F-CiMAM DNA fragment. The CiMAM-F-CiMAM DNA fragment was cut by *Age*I and *Xba*I and ligated into the corresponding cleavage sties of 6LFP-deleted plasmid pP43-6LFB-GFP to construct the 6206-bp plasmid P43-2CiMAM-GFP. As shown in Figure 1, pP43-2CiMAM-GFP contained two copies of CiMAM-A24 cDNAs fused with GFP reporter (CiMAM- F-CiMAM-GFP) and driven by P43 promoter of *B. subtilis.* When the resultant plasmid pP43-2CiMAM-GFP was restricted with *Age*I and *Xba*I, a 2CiMAM fragment with a molecular size of about 200 bp and a backbone with 6 kb was obtained (Figure 1C).

### 2.2. Gene Transfer and Screening 

Each time, 1.3 × 10^8^ competent cells were applied for gene transfer, resulting in 15 to 30 colonies grown on tetracycline-selective agar plates. The average transfer rate was 1.1 × 10^−5^~2.3 × 10^−6^%. Finally, we obtained about 351 putative colonies for further PCR screening.

Next, we used PCR to detect whether the GFP reporter cDNA was presented in the genomic DNAs extracted from those colonies. A total of about 170 transformants was found. Subsequently, we used Western blot to detect whether the recombinant protein was produced by these transformants using GFP antibody. Forty transgenic strains clearly exhibited a positive signal. Finally, we employed an agar well diffusion assay to detect whether antimicrobial capability was stably present in the extract of transformants for which an inhibition zone was shown on agar plate confluent with test microorganism *S. epidermidis*. Two transgenic strains, C117 and C166, were finally selected for further study since they presented the highest antibacterial activity after several passages. 

### 2.3. Using PCR to Detect the Exogenous Plasmid Transferred into B. subtilis Host Cells 

Plasmid stability was evaluated as the ability to maintain antibacterial activity after continuously culturing transgenic strains C117 and C166 for 10 passages. Their DNAs were extracted and used as templates for PCR analysis to amplify the *GFP* cDNA within plasmid P43-2CiMAM-GFP with forward primer GFP-*Nhe*I and reverse primer GFP-*Hin*dIII (Figure 1B). No DNA fragment was amplified from *B. subtilis* WB800 which served as negative control (Figure 1D). However, a DNA fragment of about 700-bp was amplified from the DNAs extracted from transgenic strains C117 and C166. This PCR product was the same in molecular size as that amplified from the *GFP* cDNA within plasmid P43-2CiMAM-GFP, which served as positive control (Figure 1D), suggesting that the transgenic strains C117 and C166 harbored plasmid P43-2CiMAM-GFP after gene transfer. 

### 2.4. Copy Number of Plasmid Transferred to the Transgenic B. subtilis Strains

Plasmid copy numbers isolated from non-transgenic WB800 and transgenic strains C117 and C166 were quantified when the absorbance optical density (O.D.) of bacterial strains grown in liquid broth reached 1, which isolated from 1 ml contained the cell density of 8.7 × 10^10^, 1.45 × 10^8^ and 2.17 × 10^8^ colony forming unit (CFU), respectively. Using VisionWorks software, we demonstrated that 20 ng of plasmid DNA were equivalent to 4.28 × 10^4^ units, 10 ng of plasmid DNA were equivalent to 3.6 × 10^4^ units, and 5 ng of plasmid DNA were equivalent to 2.29 × 10^4^ units (Figure 2B). The signal intensities obtained from these three known amount of plasmid DNAs were then converted into the trendline, resulting in the linear regression equation of *y* = 1238.7*x* + 19,457 with *R*^2^ equal to 0.8684 (Figure 2C).

In a parallel experiment, plasmid DNAs extracted from 10, 5, and 2.5 μL cultured solution of *B. subtilis* C117 strain exhibited 3.15 × 10^4^, 2.1 × 10^4^ and 1.02 × 10^4^ unit, respectively, while plasmid DNAs extracted from 10, 5 and 2.5 μL cultured solution of C166 strain exhibited 2.54 × 10^4^, 1.84 × 10^4^ and 9.63 × 10^3^ unit, respectively (Figure 2B). Signal intensity of the plasmid extracted from 10 μL of cultured transgenic C117 and C166 strains was located within the range of the linear regression equation. Therefore, we substituted 3.15 × 10^4^ and 2.54 × 10^4^ unit into the equation, resulting in plasmid amount of 9.76 and 4.82 ng, respectively. Meanwhile, the molecular weight of plasmid pP43-2CiMAM-GFP was 3.83 × 10^3^ kDa, and the transgenic C117 and C166 strains contained 1.45 × 10^6^ and 2.17 × 10^6^ CFU (10 μL), respectively. Based on these parameters, the plasmid copy numbers of transgenic *B. subtilis* C117 and C166 strains were 1057 and 349 copies per cell, respectively. 

### 2.5. Western Blot Analysis to Prove That Recombinant Protein Containing CiMAM Was Produced by the Transgenic B. subtilis Strains

Using Coomassie blue staining after SDS-PAGE, we found that an extra band with molecular weight of 37 kDa was only exhibited in the protein patterns of the two transgenic strains, but absent in the protein pattern shown in the non-transgenic *B. subtilis* WT800 strain (Figure 3A). Additionally, based on Western blot analysis, no positive signal could be detected in the negative control (N), a non-transgenic *B. subtilis* WB800 strain (Figure 3B). In contrast, a positive band located at the approximate molecular weight of 37 kDa was shown in the proteins extracted from the transgenic C117 and C166 strains after hybridization. The molecular weight of this positive band corresponded to that of recombinant protein CiMAM-F-CiMAM fused with GFP reporter. 

### 2.6. Using Amino Acid Sequencing to Further Confirm That Recombinant Proteins Were Produced by Transgenic Strains

To further confirm the above finding, we excised the extra band with molecular weight of 37 kDa shown on SDS-PAGE after Coomassie blue staining, followed by Liquid Chromatography Tandem Mass Spectrometry (LC-MS/MS). LC-MS/MS detected 51% of the amino acid residues of CiMAM-GFP fusion protein from the recombinant protein produced by transgenic C117, in which 17% coverage was located at CiMAM and 75% at GFP. Similarly, 45% of the amino acid residues of CiMAM-GFP fusion protein could be detected from the recombinant protein produced by transgenic C166, in which 17% coverage was located at CiMAM and 65% at GFP. This evidence supported that the 37-kDa protein shown on transgenic *B. subtilis* C117 and C166 strains is a recombinant CiMAM fused GFP protein. Taken together, we conclude that transgenic C117 and C166 strains can produce recombinant protein CiMAM-F-CiMAM-GFP encoded by the plasmid P43-2CiMAM-GFP we transferred.

### 2.7. Bactericidal Activity of the Extracts from Transgenic B. subtilis Strains against Halophilic Pathogens 

Since CiMAM AMP originates from marine cholate, we first determined whether the recombinant fusion protein CiMAM-F-CiMAM-GFP-containing *B. subtilis* strains exhibit specific antimicrobial activity against halophilic pathogens, such as *V. parahaemolyticus*, *V. alginolyticus* and *V. natriegens*. To make this determination, an agar well diffusion assay was used to analyze bactericidal activity. After the agar plates were incubated at 37 °C for 8 h, the well containing Tetracycline exhibited an inhibition zone and thus served as positive control. However, the inhibition zone was neither observed as the extract from non-transgenic *B. subtilis* WB800, which served as the first negative control, nor was it observed as the extract from the transgenic *B. subtilis* C117 strain without pepsin digestion (Figure 4A,D,G), which served as the second negative control (C117’). 

To study antimicrobial activity against *V. parahaemolyticus*, it was first necessary to assess the effect of different dosages of Tetracycline against *V. parahaemolyticus.* Tetracycline with dosages of 0.5, 1, and 1.50 μg displayed an inhibition zone averaging 5.7, 6.82 and 7.34 mm in size, respectively (Figure 4A,B). The relationship between Tetracycline dosage and the resultant size of inhibition zone was converted into the trendline to obtain the linear regression equation, as *y* = 1.6417*x* + 4.9794 with *R*^2^ equal to 0.9585 (Figure 4C). In the parallel experiment, we observed that the extracts from transgenic *B. subtilis* C117 and C166 strains displayed inhibition zones with average size of 6.53 and 5.8 mm, respectively (Figure 4A,B), suggesting that the recombinant CiMAM-containing extract treated with pepsin possesses bactericidal activity. After substituting the size of inhibition zones obtained from extracts of transgenic strains C117 and C166 into the linear regression equation, we discovered antibacterial potency against *V. parahaemolyticus* equivalent to 0.94 and 0.5 μg of Tetracycline, respectively. Since the examined extracts were obtained from bacterial strains grown at the OD_600_ = 1, the cell numbers of C117 and C166 strains were 2.9 × 10^6^ and 4.34 × 10^6^ CFU, respectively. The total amount of extract loaded into each well to perform the agar well diffusion assay was 20 μL. Therefore, we calculated that the bactericidal activity of 1 μL out of 5.5 mL extract from transgenic *B. subtilis* C117 and C166 strains against *V. parahaemolyticus* was approximately equivalent to the potency of 47.07 and 25.2 ng of Tetracycline, respectively. 

To study antimicrobial activity against *V. alginolyticus*, we demonstrated that the dosages of 0.25, 0.75, and 1.50 ng of Tetracycline exhibited an average inhibition zone of 3.85, 5.83, and 6.92 mm, respectively (Figure 4D,E). As described above, the linear regression equation was obtained (Figure 4F and Table 1). The extracts from transgenic *B. subtilis* C117 and C166 strains displayed an average inhibition zone of 6.32 and 5.29 mm, respectively (Figure 4D,E). After substituting the size of inhibition zones obtained from extracts of transgenic strains C117 and C166 into the linear regression equation, we discovered antibacterial potency against *V. alginolyticus* equivalent to 1.16 and 0.73 μg of Tetracycline, respectively. In accordance with the procedure described above, we calculated that the bactericidal activity of 1 μL out of 5.5 mL extract from transgenic *B. subtilis* C117 and C166 strains against *V. alginolyticus* was approximately equivalent to the potency of 58.17 and 36.55 ng of Tetracycline, respectively. 

Last, we examined the inhibitory effect of transgenic *B. subtilis* extract on *V. natriegens*. The results showed that different dosages of Tetracycline with dosages of 0.5, 1 and 1.50 μg displayed an average inhibition zone of 5.77, 6.71, and 7.33 mm, respectively (Figure 4G,H). Since the linear regression equation was obtained (Figure 4I and Table 1), we obtained an antibacterial potency against *V. natriegens* (Figure 4I) equivalent to 1.14 and 0.65 μg of Tetracycline. In accordance with the procedure described above, we calculated that the bactericidal activity of 1 μL extract from transgenic *B. subtilis* C117 and C166 strains against *V. natriegens* was approximately equivalent to the potency of 57.06 and 32.35 ng of Tetracycline, respectively.

Interestingly, we went further to compare antibacterial activity between the recombinant CiMAM-expressing transgenic *B. subtilis* C117 strain and lactoferricin-expressing transgenic *B. subtilis* T13 strain reported by Lee et al. (2019). When the cell lysates extracted from these two strains were added into the well on a 2.5%-NaCl-containing agar plate confluent with the halophilic pathogen *V. natriegens,* we found that the bactericidal effect of transgenic *B. subtilis* C117 on *V. natriegens* was higher than that of the transgenic T13 strain (Figure 4J,K).

### 2.8. Bactericidal Activity of Recombinant CiMAM-Containing Extracts from Transgenic Strains C117 and C166 against the Euryhaline Pathogens 

Similar to the strategy of using the agar well diffusion assay, we carried out another antimicrobial study of recombinant CiMAM-containing extracts from transgenic strains on the common euryhaline pathogens *E. tarda* and *S. iniae*. As shown in the Appendix A, the extract from *B. subtilis* WB800 and from transgenic C117 strain without pepsin treatment did not exhibit an inhibition zone and, thus, served as negative control. After measuring the size of inhibition zone on the *E. tarda*-confluent agar plate, we observed that Ampicillin with dosages of 0.25, 0.75, and 1.5 μg displayed an average inhibition zone of 2.25, 3.9, and 5.1 mm, respectively. Thus, the linear regression equation was obtained (Appendix A and Table 1). The extracts of transgenic *B. subtilis* C117 and C166 strains displayed an average inhibition zone of 3.61 and 3.56 mm, respectively (Appendix A). Therefore, we determined antibacterial potency against *E. tarda* equivalent to 0.77 and 0.75 μg of Ampicillin, respectively. The bactericidal activity against *E. tarda* of 1 μL of transgenic *B. subtilis* C117 strain was approximately equivalent to the potency of 38.59 ng of Ampicillin, while that of transgenic *B. subtilis* C166 strain was equivalent to 37.47 ng of Ampicillin. 

Meanwhile, in the bactericidal activity study of *S. iniae*, as shown in the Appendix A, Tetracycline served as a positive control, and it was found that dosages of Tetracycline with dosages of 0.50, 1.00, and 1.50 μg exhibited an average inhibition zone of 4.87, 5.61 and 6.58 mm, respectively. Thus, the linear regression equation was obtained (Appendix A and Table 1). In the parallel experiment, we observed that extracts of the transgenic *B. subtilis* C117 and C166 strains displayed average inhibition zones of 5.71 and 5.32 mm, respectively (Appendix A). Therefore, we determined that the antibacterial potency of transgenic C117 and C166 strains against *S. iniae* were equivalent to 1.05 and 0.79 μg of Tetracycline, respectively. Similarly, following the calculation described above, we concluded that the antimicrobial activity of extracts obtained from 5 mL of C117 and C166 bacterial liquid solution against *S. iniae* was equivalent to 50.97 ng of Tetracycline per 1 μL of C117 extract and 39.28 ng of Tetracycline per 1 μL of C166 extract.

### 2.9. Bactericidal Activity of Recombinant CiMAM-Containing Extracts from Transgenic Strains C117 and C166 against Common Freshwater Pathogens

Next, we determined the bactericidal activity of transgenic strains against common freshwater pathogen, such as Gam-positive *S. epidermis.* As shown in the Appendix A, the wells on agar plates containing Ampicillin again exhibited an inhibition zone and thus served as positive control, while the extracts from *B. subtilis* WB800 and from transgenic C117 strain without pepsin treatment, which did not exhibit an inhibition zone, served as negative controls. After measuring the size of inhibition zone on the *S. epidermidis*-confluent agar plate, we determined that Ampicillin with a dosage of 0.5, 1.5, and 2 μg displayed an average inhibition zone of 1.5, 3.14, and 3.75 mm, respectively. Meanwhile, extracts of the transgenic *B. subtilis* C117 and C166 strains displayed average inhibition zones of 2.23 and 2.00 mm, respectively (Appendix A). Thus, based on the linear regression equation (Appendix A and Table 1), we determined antibacterial potency against *S. epidermidis* with equivalency of 0.96 and 0.81 ng of Ampicillin, respectively. Therefore, the bactericidal activity against *S. epidermidis* of 1 μL of transgenic *B. subtilis* C117 strain was approximately equivalent to the potency of 47.97 ng of Ampicillin, while that of transgenic *B. subtilis* C166 strain was equivalent to 40.5 ng of Ampicillin.

## 3. Discussion

### 3.1. Recombinant CiMAM -Expressing Transgenic B. subtilis Exhibits Bactericidal Activity against a Variety of Bacteria

It has been reported that CiMAM, an antibacterial peptide isolated from one of the marine cholates, exhibits a wide range of antimicrobial effects against Gram-positive bacteria, such as *Bacillus megaterium*, *B. subtilis*, *Staphylococcus aureus*, *S. epidermidis* and *Planococcus citreus* and Gram-negative bacteria, such as *Pseudomonas aeruginosa*, *V. alginolyticus*, *Listonella anguillarum*, *Yersinia enterocolitica*, *Serratia marcescens*, *Klebsiella pneumoniae* and *E. coli* K-12 strain D31, as well as fungi, such as *Candida albicans* [7]. In this study, the pepsin-digested extracts from recombinant CiMAM-expressing transgenic *B. subtilis* strains displayed bactericidal activity against *S. epidermidis* and *V. alginolyticus,* consistent with the results of Fedders et al. [7]. We also demonstrated that these CiMAM-expressing *B. subtilis* strains possessed bactericidal activity against six more bacterial species, including Gram-positive *S. epidermidis* and *S. iniae* and Gram-negative *E. tarda*, *V. alginolyticus*, *V. natriegens,* and *V. parahaemolyticus*.

We categorized the nine species of pathogens examined in this study into three different degrees of salt concentration for bacterial growth. For example, *S. epidermidis* grow in freshwater, *E. tarda* and *S. iniae* grow in euryhaline seawater, and *V. alginolyticus*, *V. natriegens* and *V. parahaemolyticus* grow in halophilic seawater. Therefore, we concluded that the bactericidal activity driven by the pepsin-digested extract from the recombinant CiMAM-expressing *B. subtilis* strains covers a broad spectrum of bacterial pathogens that cause fish and shellfish diseases.

Nevertheless, we noticed that CiMAM also has antibacterial activity against *B. subtilis* [7], a host cell used for gene transfer. Therefore, to prevent the transgenic strains from being attacked by CiMAM produced in the *B. subtilis* transformants, we constructed an expression plasmid in which double repeats of CiMAM cDNAs, separated by a cleavage site of pepsin, was designed. The recombinant protein translated from this construct was a molecule of CiMAM, and finally formulated as an intact CiMAM-F-CiMAM-GFP recombinant fusion protein, which was completely lacking in antibacterial activity. Such construct was also driven by the p43 promoter, and, indeed, such hypothesis was borne out by the evidence from our agar well diffusion assay, as shown in Figure 4 and Appendix A, where we demonstrated no antimicrobial effect if the extracts obtained from the CiMAM-expressing transgenic *B. subtilis* strains were not treated with pepsin, as shown in the C117’ negative control group. Therefore, we concluded that *B. subtilis* transgenic strains were not killed by the original CiMAM-F-CiMAM-GFP recombinant fusion protein unless treated with pepsin. However, once the recombinant CiMAM-expressing transgenic *B. subtilis* strains were fed to fish and farm animals, the original CiMAM-F-CiMAM-GFP recombinant fusion protein would be cleaved by pepsin in the stomach, releasing a single functional CiMAM molecule exhibiting broad antimicrobial activity and, hence, might have high potential for application in the animal husbandry and aquaculture farming industries. 

### 3.2. The Effect of Salinity on the Efficacy of Antimicrobial Peptides

Since CiMAM is derived from a marine organism, it would have antibacterial activity under high salinity, such as that reported for *P. citreus, Listonella anguillarum,* and *V. alginolyticus* [7]. In the present study, the extract from transgenic *B. subtilis* strains expressing the CiMAM recombinant protein showed bactericidal effect against halophilic bacteria, such as *V. alginolyticus*, *V. natriegens*, and *V. parahaemolyticus,* suggesting that the antimicrobial activity of CiMAM is not diminished in a highly saline environment, not a common feature of AMPs. In contrast, the antimicrobial activity of lactoferricin from cattle milk declines faster under a high-salt environment. CiMAM maintains its antimicrobial activity in NaCl solution at the concentration of 450 mM [7], while the antimicrobial activity of lactoferricin decreases drastically in NaCl solution at the concentration of 100 mM [17]. Especially, the antimicrobial activity of lactoferricin also decreases significantly in the presence of MgCl_2_ or CaCl_2_ over 5 mM in the environment [17]. Lee et al. [16] demonstrated that the extracts obtained from lactoferricin-expressing transgenic *B. subtilis* strains T1 and T13 exhibit bactericidal activity against *V. parahaemolyticus*. Transgenic T1 and T13 strains employed the P43 promoter to express recombinant lactoferricin; the transgenic C117 strain employs the same promoter to express recombinant CiMAM. The copy number of harbored plasmid is also similar among T1, T13, and C117 strains. However, compared to transgenic T1 and T13 strains, the antimicrobial capability of transgenic C117 in this study exhibits greater efficacy against *V. parahaemolyticus* (Table 2). Moreover, the bactericidal effect of transgenic *B. subtilis* C117 on *V. natriegens* was higher than that of the transgenic T13 strain (Figure 4J,K). In addition to NaCl, natural seawater also contains 479 mM of sodium ions, 54.5 mM of magnesium ions, and 10.5 mM of calcium ions [18], which are likely to influence the activity of AMPs like lactoferricin, but not CiMAM.

### 3.3. Recombinant Fusion Protein CiMAM-F-CiMAM-GFP Expressed in Transgenic B. subtilis Strains

The recombinant CiMAM expression construct used for transformation is 6.2 kDa, in which the main DNA sequence encoding CiMAM recombinant protein (37 kDa) includes a YncM (4.5 kDa) secretion signal fragment, a double-repeated CiMAM cDNA (2 × 3.25 kDa) and a reporter GFP cDNA (26 kDa). The expected molecular weight of this expression plasmid is around 37 kDa. However, when the total proteins extracted from transgenic strains were analyzed by Western blot using antibody against GFP, three major positive bands were shown: a band of 37 kDa, which is the expected molecular size of recombinant CiMAM protein, and two bands near 26 and 48 kDa, which were unexpected proteins. Since the molecular size of reporter GFP cDNA is 26 kDa, the 26 kDa-positive band should be the reporter GFP, resulting from the natural cleavage of recombinant CiMAM protein. We speculate that the 48 kDa-positive protein fragment might be an incomplete dimer form of CiMAM recombinant protein comprised of one complete form of CiMAM recombinant protein (37 kDa) and one truncated form (11 kDa) of it. This truncated form of CiMAM recombinant protein is, in turn, comprised of a YncM secretion signal fragment (4.5 kDa) and a double-repeated CiMAM fragment (2 × 3.25 kDa) without containing GFP reporter. 

CiMAM consists of a single α-helix with a hydrophilic zone and a hydrophobic zone formed on both sides of the arc of cylindrical structure [7]. Thus, both complete and truncated (i.e., without GFP fragment) forms of CiMAM recombinant protein compose a four α-helix structure which is a more stable structure, as described by Chou et al. [19]. This four-α-helix CiMAM mixture could be anchored to each other by the hydrophobic region to form a more robust polymer structure generating the 48 kDa signal, as observed in Western blot analysis. Finally, to determine the authentic composition of this 48-kDa band, we sequenced the protein of this size. Although the presence of GFP was confirmed, no intact CiMAM sequence was identified, calling for further study to identify this recombinant protein. Finally, we suggest that the host *B. subtilis* WB800 strain may still contain endogenous proteases which somehow favor cleavage of the recombinant fusion protein CiMAM-F-CiMAM-GFP, even though it has eight missing proteases, as reported by Wu et al. [20].

### 3.4. The GFP Fragment Cleaved from Recombinant Fusion Protein CiMAM-F-CiMAM-GFP Served as a Selection Marker for Screening Transformants

The GFP cDNA contained in the expression vectors of pP43-6LFB-GFP [16] was constructed to serve as a selection marker to screen transgenic colonies. Unexpectedly, when plasmid pP43-6LFB-GFP was introduced into *B. subtilis*, the green fluorescent signal was barely observed in the lactoferricin-expressing transgenic strains under the fluorescence microscope [16]. In contrast, when plasmid pP43-2CiMAM-GFP was introduced into *B. subtilis* in this study, the green fluorescent signal was observed in the recombinant CiMAM-expressing transgenic colonies after they were cultured on a plate for seven days at room temperature (Appendix A). We reasoned that the conflicting outcome may have resulted from structural differences between recombinant lactoferricin and CiMAM produced by the transgenic *B. subtilis* strains. Based on the Western blot analysis, three positive bands appeared among total proteins extracted from recombinant CiMAM-expressing transgenic strains, as shown in Figure 3, while only a single positive band was observed among total proteins extracted from lactoferricin-expressing transgenic strains, as reported by Lee et al. [16]. As previously described, one of the three positive bands was the monomeric 26 kDa-GFP, resulting from the cleavage of dimer recombinant CiMAM proteins. Thus, the green fluorescence signal was clearly visible in the transgenic colonies. Nevertheless, the molecular size of the single band of transgenic T1 and T13 corresponds to a monomer of recombinant lactoferricin protein, suggesting that the GFP peptide is not released from monomeric recombinant lactoferricin. Instead, it is likely that the GFP motif is masked, or hindered, by the entire structure of recombinant lactoferricin such that the green fluorescence signal cannot be emitted. It is the GFP cleaved from dimeric recombinant CiMAM protein that produces the green fluorescence signal in recombinant CiMAM-expressing transgenic *B. subtilis* strains. Therefore, compared to using the antibiotic selection method, the green fluorescence approach to screen the putative colonies, which might express the recombinant CiMAM protein, is more effective. For example, only 30% of colonies grown on the tetracycline-containing agar plate exhibited green fluorescence. Consequently, the tedious screening work can be greatly reduced in terms of time, effort, and cost.

### 3.5. Correlation between Plasmid Copy Number and Antimicrobial Ability of Transgenic B. subtilis Strain

While screening transformants, we found that the resultant transformants presented different degrees of antimicrobial efficacy, even though all host *B. subtilis* cells were transformed with the same expression plasmid. Through dot blot analysis, we could clearly determine the copy number of plasmid harbored by each transformant. For example, transgenic *B. subtilis* C117 and C166 strains contained plasmids with 1057 and 349 copies per cell, respectively. On the other hand, we found that the C117 strain displays higher antimicrobial activity than the C166 strain by having a higher copy number of plasmid, suggesting that plasmid copy number harbored by transformants is a critical factor affecting the antimicrobial efficacy of the transgenic *B. subtilis* strains. Although the plasmid copy number does not exhibit an absolute linear correlation with the antimicrobial efficacy of the transgenic *B. subtilis* strains, the number of plasmids harbored by transformants is still an important factor affecting antimicrobial efficacy among the transgenic *B. subtilis* strains, such as transgenic C117 versus C166 and C117 versus T13. However, other factors may be at play, such as the physiological condition and the ability of translational modification of the recombinant protein within transformants. 

## 4. Materials and Methods

### 4.1. Bacterial Culture Condition

The host cell used in this study was *B. subtilis* WB800 strain [21], which was a gift from Dr. Sui-Lam Wong, University of Calgary, Canada. It contains the deletion of eight extracellular proteases, including *nprE*, *aprE*, *epr*, *bpr*, *mpr*, *nprB*, *Δvpr,* and *wprA* [20]. The pathogens used as targets for bactericidal testing in an agar well diffusion assay were *Edwardsiella tarda* (BCRC 16702), *Staphylococcus epidermidis* (BCRC 15245), *Streptococcus iniae*, *Vibrio parahaemolyticus* (BCRC 12866), *V. natriegens* (BCRC 10805), and *V. alginolyticus*. They were all purchased from the Bioresource Collection and Research Center (BCRC), *E. tarda* were cultured in Luria-Bertani (LB) broth medium, *S. epidermidis* in nutrient broth medium, *V. parahaemolyticus* and *V. alginolyticus* in Tryptic soy medium (1.5% Tryptone (BD 211705), 0.5% Soytone (BD 243620)) supplemented with 2.5% NaCl (UniRegion UR-SC001). *S. iniae* was cultured in Todd Hewitt broth medium (SIGMA). Specifically, we used tryptic soy medium mixed with 3% fiber sheep blood to perform agar well diffusion assay for *S. iniae*. All of the above bacteria were incubated at 37 °C for 18 h, except *S. iniae*, which was incubated for 24 h.

### 4.2. Construction of Expression Plasmid 

According to the cDNA sequence of CiMAM reported by Fedders et al. (2008), two sets of primers were designed to synthesize a template for cloning a double-repeated CiMAM cDNA interrupted with Phenylalanine codon in between (CiMAM-F-CiMAM). As illustrated in Figure 1A, one set of primers for cloning the 5’-end of cDNA fragment was sense primer (TCTAGAGCTAGCATGTTTTGGGGCTCAAGAAGAGCACTGCCGAAACTGGCACACTCACTGAGAC) and antisense primer (TGAGCCCCAAAACCATCTTGACAGGCCTCTTGTCAGCAGTCTCAGTGAGTGTGCCAGTT) which were used to react in a volume of 50 μL under the following condition for 25 cycles: denaturation at 95 °C for 30 s, annealing at 60 °C for 30 s, and extension at 72 °C for 90 s. This PCR strategy was to be repeated using another set of primers for cloning the 3’-end of cDNA fragment, which was sense primer (CAAGATGGTTTTGGGGCTCAAGAAGAGCACTGCCGAAACTGGCACACTCACTGAGACT) and antisense primer (ACCGGTGCTAGCTTAAACATCTTGACAGGCCTCTTGTCAGCAGTCTCAGTGAGTGTGCCA). After finishing PCR, 5 μL was taken from each of the above two set PCR resultant solutions served as a template for performing another round of PCR using the sense primer (TCTAGAGCTAGCATGTTTTGGGGCTCAAGAAGAGCACTGCCGAAACTGGCACACTCACTGAGAC) and antisense primer (ACCGGTGCTAGCTTAAACCATCTTGACAGGCCTCTTGTCAGCAGTCTCAGTGAGTGTGCCA). The PCR condition for amplifying the final product with the following condition was: 1 cycle of denaturation at 98 °C for 5 min, annealing at 42 °C for 1 min, and extension at 72 °C for 90 s, followed by 35 cycles of denaturation at 98 °C for 30 s, annealing at 60 °C for 30 s, and extension at 72 °C for 90 s. 

After digestion by *Age*I and *Xba*I, six-repeated lactoferricin cDNA was removed from plasmid pP43-6LFB-GFP [16] and replaced by the above final PCR product cut with corresponding enzymes to generate plasmid p43-2CiMAM-GFP. This expression plasmid contains a p43 promoter, a GFP reporter, a tandem repeat of CiMAM cDNA derived from *Ciona intestinalis*, and two antibiotic-resistant genes.

The recombinant fusion protein containing CiMAM designed in this study consisted of two copies of CiMAM separated by phenylalanine (F) and fused with GFP reporter at the C-terminus (CiMAM-F-CiMAM-GFP). This became the inactive bactericidal form. However, once the CiMAM-F-CiMAM-GFP recombinant protein was digested by pepsin at F residue, two molecules of the active bactericidal form of CiMAM were released.

### 4.3. Preparation of Competent Cells, Electroporation, and PCR Detection

All these methods were described previously by Lee et al. [16], except that plasmid pP43-2CiMAM-GFP (50 μg/mL) was performed 12 times electroporation and each time, there were 1.3 × 10^8^ competent cells applied for gene transfer.

### 4.4. Dot Blot Analysis and Copy Number of Plasmids

To identify the copy number of plasmid p43-2CiMAM-GFP transferred to transgenic *B. subtilis* strains, we employed a Digoxigenin-labeled GFP DNA fragment as a probe to hybridize with the plasmid isolated from transgenic strains using dot blot analysis. When the OD_600_ of *B. subtilis* strains reached 1, we extracted their plasmids following the instructions provided with the Plasmid Extraction kit (Arrowtec). After extraction, 0.5 μL of plasmid DNA sample was dropped and fixed on a nylon-cellulose (NC) transfer membrane (Millipore, Burlington, MA, USA) by ultraviolet light (Ultra-Lum UVC-515) under 120 millijoules five times. Then, the NC membrane was immersed in pre-hybridization solution (50% formamide, 5× Saline-Sodium Citrate (SSC) buffer, 3× Denhardt’s Solution (Sigma, St. Louis, MO, USA), 200 μg/mL salmon sperm, and 0.1% SDS) at 37 °C for 2 h. Afterwards, the pre-hybridization solution was replaced by a fresh hybridization solution containing GFP probe labeled with Digoxigenin prepared, as described previously [16], followed by reaction at 42 °C for 16 h. After hybridization, the NC membrane was washed with 2 × SSC at 37 °C for 15 min, followed by continuous washing with 1 × SSC at 42 °C for another 15 min. Finally, 0.1 × SSC was used to wash NC membrane at 68 °C for 1 h, and then the membrane was blocked with a blocking solution (Phosphate-buffered saline (PBS), 5% sheep serum, 1% Bovine serum albumin) for 3 h, followed by blotting with 1:8000 Anti-Digoxigenin antibody (Roche, Nutley, NJ, USA) at 4 °C for 16 h. After reaction, the membrane was washed six times with the mixture of PBS and 0.1% Tween 20 for 15 min, followed by washing three times with Tris-MgCl2-NaCl (TMN) buffer at pH 9.5 [100 mM Tris base (Uniregion, Taipei, Taiwan), 50 mM MgCl_2_ (J.T. Baker, Phillipsburg, NJ, USA), 100 mM NaCl (Uniregion, Taipei, Taiwan), and 0.1% Tween-20 (Sigma, St. Louis, MO, USA), pH9.5 [22] for 5 min. Subsequently, TMN buffer supplemented with 3.5 μL 5-Bromo-4-chloro-3-indolyl phosphate (50 μg·μL^−1^; Roche) and 4.5 μL Nitro Blue Tetrazolium (100 μg·μL^−1^) (Roche, Nutley, NJ, USA) was further employed per ml, followed by color reaction at 4 °C for 1 to 2 h. The resultant signals were recorded by luminometer (ChemiDoc-it 815, Bio-Rad, Hercules, CA, USA), and the images were analyzed by VisonWorks software (version 6.7.4, Upland, CA, USA), in which The Area Density Function was used to identify the signal position and calculate the color intensity. Three points of positive control were used to make an xy scatter diagram together with the mass weight of the DNA. Mass weight of DNA and mean density were taken as the x- and y-axes, respectively. The relationship between the x- and y-axes was converted into the trendline to obtain the Linear Regression Equation. The plasmid copy number was calculated as plasmid copy number = M/MW × V × CFU × N_A_, where M is plasmid mass weight (g), MW is plasmid molecular weight, V is volume of bacterial culture solution (mL), CFU is colony-forming unit per ml at OD_600_ = 1, and N_A_ is the Avogadro Constant. The DNAs from purified plasmid p43-2CiMAM-GFP and from non-transgenic *B. subtilis* WB800 strain served as positive and negative control, respectively.

### 4.5. Protein Analysis by SDS-PAGE

We used Coomassie blue staining to present the protein profiles of total proteins extracted from the wild-type and transgenic *B. subtilis* strains. After *B. subtilis* strains were cultured in 3 mL LB at 37 °C for 16 h, each bacterial sample was mixed with sample buffer at a ratio of 4 to 1. The mixture was heated at 100 °C for 6 min and then placed on the ice for 5 min. The total proteins extracted from each sample were subjected to SDS-PAGE on a 12% polyacrylamide gel at 80V for 30 min, followed by adjusting the voltage to 120 V for 80 min. After electrophoresis, the gel was stained with Coomassie blue [0.1% Coomassie Brilliant Blue (Sigma, St. Louis, MO, USA), 50% Methanol, 10% acetic acid glacial (Thermo Fisher, Waltham, MA, USA)] for 2 h and immersed in de-staining buffer (40% Methanol, 10% acetic acid glacial) overnight. 

### 4.6. Western Blot Analysis

Western blot analysis was performed to confirm whether transgenic strains could express recombinant protein CiMAM-F-CiMAM fused with GFP reporter, using polyclonal antibody against GFP as a probe. After total proteins were analyzed on a 12% polyacrylamide gel under 100 voltage with 400 mA for 80 min, the proteins were transferred to a PVDF membrane blocked with blocking solution [5% skimmed milk powder, 300 mM NaCl (Uniregion), 40 mM Tris base (Uniregion), 0.1% Tween-20 (Sigma, St. Louis, MO, USA)] for 1 h at room temperature, followed by immunoblotting with a primary rabbit-anti-GFP antibody [16] with dilution of 1:2000 at 4 °C overnight. After reaction, the gel was washed four times with the mixture of Tris-buffered saline and 0.1% Tween 20 (TBST) for 5 min each time. Secondary antibody of goat-anti-rabbit-HRT (Santa Cruz, Dallas, TX, USA) with the dilution 1:5000 was reacted for 1 h at room temperature, followed by the washing procedure described above. Finally, ECL (Millipore WBKLS0500) was used for luminescence color reaction, and the results were recorded by a luminescence spectrometer (ChemiDoc-it 815, Bio-Rad, Hercules, CA, USA).

### 4.7. Peptide Sequence of the CiMAM-F-CiMAM-GFP Recombinant Fusion Protein by LC-MS/MS

The total proteins were extracted from one-ml of transgenic strains C117 and C166 grown at OD_600_ = 1. After centrifugation, the extracts were resuspended in 60 μL 1 × sample buffer (10% SDS, 50% glycerol, 0.05 M DTT, 0.01 M EDTA, 0.05% bromophenol blue and 0.125 M Tris-HCl, pH 6.8; GeneMark, Taichung, Taiwan). Then, 20 μL of supernatant were subjected to SDS-PAGE and stained with Coomassie Blue. After de-staining, the protein band at the position of molecular weight 37 kDa was cut from the gel. The gel of each sample was cut into 1 mm^3^ small pieces, reduced with 50 mM DTT at 37 °C for 1 h, followed by alkylated with 100 mM iodoacetamide in dark at room temperature for 30 min. The gel pieces were washed with 50% acetonitrile/25 mM ammonium bicarbonate until the Coomassie blue was removed. The gel pieces were soaked in 100% acetonitrile for 5 min, dried with Speedvac, and rehydrated with 0.3 μg trypsin in 25 mM ammonium bicarbonate. Cleaned microcentrifuge pestle was used to crash the gel to perform trypsin digestion at 37 °C for 16 h. The in-gel digested samples were centrifuged, then the supernatants were transferred into a clean tube. Solution of 50% acetonitrile and 5% trifluoroacetic acid was added to the gel, and the sample was briefly sonicated to extract the tryptic peptides. Then, solution of 80% acetonitrile and 5% trifluoroacetic was used and repeated this procedure again. All the extraction solutions were combined and concentrated with Speedvac. The tryptic peptides were re-dissolved with 0.1% formic acid for following LC-MS/MS analysis of peptide sequencing (Thermo Q-Exactive LC-MS, Thermo Xcalibur 4.0, Waltham, MA, USA).

### 4.8. Bactericidal Agar Plate Assay

An agar well diffusion assay was used to determine the bactericidal activity of the extracts from the transgenic lines, following the method described previously by Lee et al. [16] with some modifications: (1) The agar plates were confluent with test pathogens, such as *V. parahaemolyticus, V. alginolyticus, V. natriegens, S. epidermidis, S. iniae,* and *E. tarda.* (2) Pathogen was cultured to OD_600_ = 0.8 prior to adding into 1.5% agar and cooling at 50 °C. (3) The extracts were suspended in 0.5 mL synthetic gastric juice [23]. (4) The extracts in a volume of 20 μL were added separately into wells of the agar plate and cultured at 37 °C for 8–10 h. (5) Size of the inhibition zone was then measured as the distance between the inhibition zone’s edge to the edge of loading well in mm. (6) Three known doses of Ampicillin/Tetracycline and their corresponding inhibition zone sizes were presented on the x–y scatter diagram, respectively. Thereafter, the linear regression equation was determined based on the linear trendline drawn on the graph. In the experimental group, 20 μL of cell extract taken from transgenic lines C117 and C166 were individually loaded on the well, followed by determination of inhibition zone size. Based on this equation, the equivalent amounts of antibiotic potency against examined pathogens from 1 μL out of 5.5 mL extracted from 6.25 × 10^8^ and 1.09 × 10^9^ CFU (at OD_600_ = 1) of transgenic *B. subtilis* C117 and C166 strains were finally obtained.

### 4.9. Comparison of Bactercidal Efficacy against the Halophilic Pathogen between Recombinant Lactoferricin and CiMAM AMPs Produced by Transgenic Strain

After 100 mL of marine broth (BD Difco^TM^, Franklin Lakes, NJ, USA) agar were sterilized, the dissolved agar was immersed in a 55 °C water bath. When cooling down, the halophilic pathogen, *V. natriegens*, grown at OD600 = 0.5, was added into the above unsolidified marine broth agar and then dispensed in plates. These agar plates contained 20‰ salinity. Next, we added the 20 μL extracts of the transgenic strain T13 expressing recombinant lactoferricin (Lee et al., 2019) and the transgenic strain C117 expressing rCiMAM individually for agar well diffusion assay. Then, we measured the inhibition zone after the agar plate with lawn confluent *V. natriegens* growth was cultured at 37 °C for 12 h.

## 5. Conclusions

CiMAM is a biodegradable antibacterial protein expressed in probiotic *B. subtilis*. Specifically, CiMAM can maintain its bactericidal activity, even in a very high salt and ion environment. Since many kinds of pathogens coexist in the cultivation environment with a wide range of salt concentration, application of recombinant CiMAM-expressing transgenic *B. subtilis* strains could reduce the use of antibiotics in the aquaculture and animal husbandry industries. Importantly, both environmental and food safety concerns for using this transgenic strain have been eliminated. 

## Figures and Tables

**Figure 1 marinedrugs-19-00111-f001:**
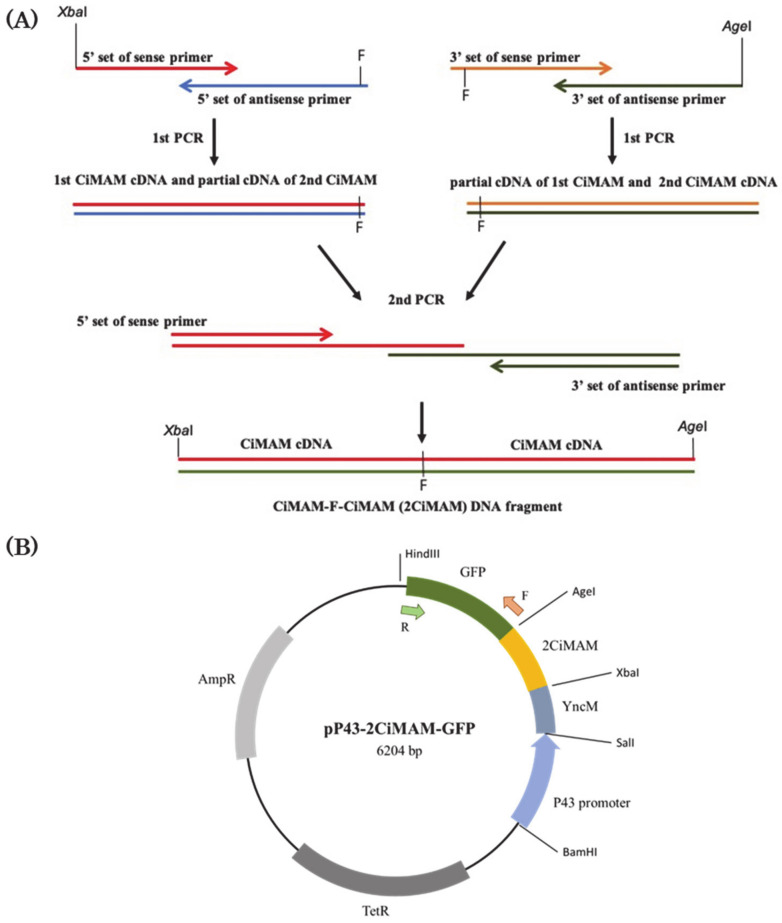
Identification and detection of plasmid pP43-2CiMAM-GFP existing in the transformed *B. subtilis*. (**A**) Strategy for cloning a CiMAM-F-CiMAM DNA fragment (2CiMAM), which contains a double-repeated CiMAM cDNA inserted with a phenylalanine (F) codon. (**B**) Schematic map of plasmid pP43-2CiMAM-GFP containing P43 promoter, signal peptide YncM cDNA sequence, 2CiMAM and GFP cDNA, which served as a selective marker. This plasmid contains anti-Tetracycline genes (TetR) and anti-Ampicillin genes (AmpR). Primers of GFP-*Nhe*I-Forward (F) and GFP-*Hin*dIII-Reverse (R) were used to amplify the GFP fragment, as indicated by arrows. (**C**) The construct expression plasmid pP43-2CiMAM-GFP was cleaved by *Age*I and *Xba*I, obtaining a 2CiMAM fragment and a backbone, as indicated by arrowhead and arrow, respectively. (**D**) Using PCR to detect the existence of plasmid pP43-2CiMAM-GFP, which served as positive control (P), while DNAs from *B. subtilis* WB800 served as negative control (N), and transgenic lines C117 and C166, as indicated, the product of PCR. M: molecular marker. The molecular mass of PCR product was 700 bp.

**Figure 2 marinedrugs-19-00111-f002:**
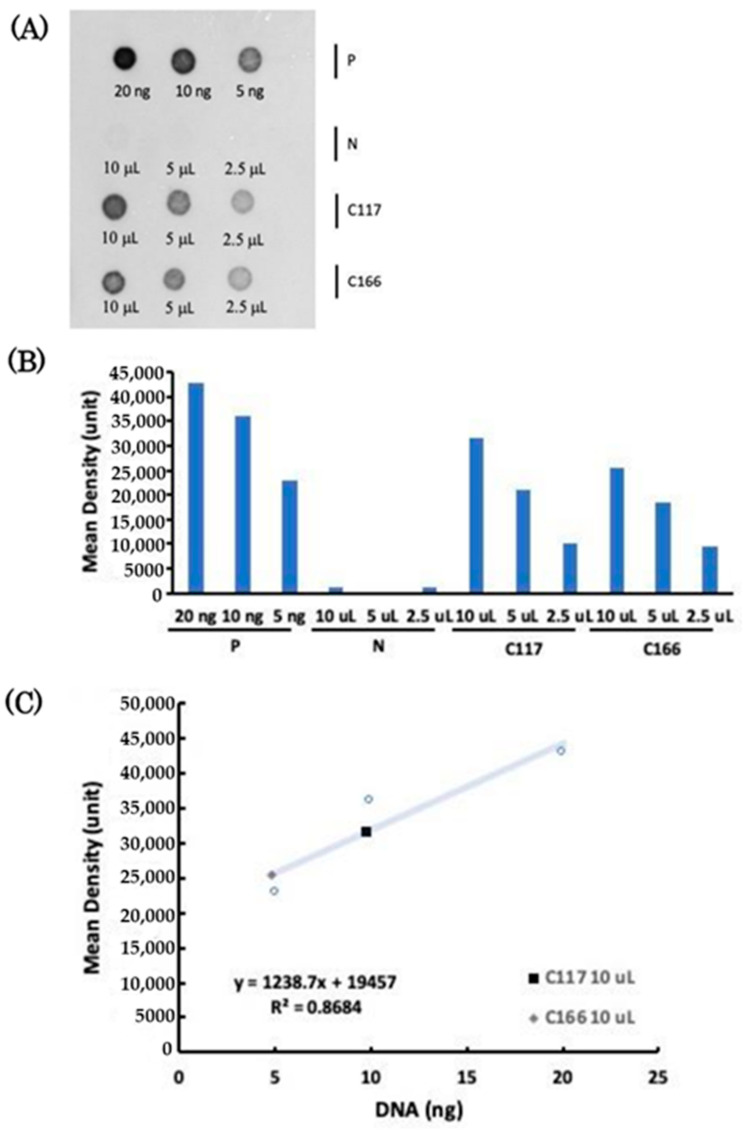
Dot blot analysis of plasmid copy number harbored by the transgenic *B. subtilis* strain. (**A**) Plasmid pP43-2CiMAM-GFP, which served as positive control (P), was presented at three different known amounts, as indicated. DNA obtained from non-transformed *B. subtilis* WB800 strain served as negative control (N), and transgenic C117 and C166 strains were examined at three different amounts of liquid-cultured bacteria, as indicated. (**B**) The signal intensities shown on (**A**) were quantified by VisionWorks software, transforming the Mean Density value (unit) into the bar chart. (**C**) The signal intensities of three known concentrations of positive control were converted into the trendline to obtain the linear regression equation.

**Figure 3 marinedrugs-19-00111-f003:**
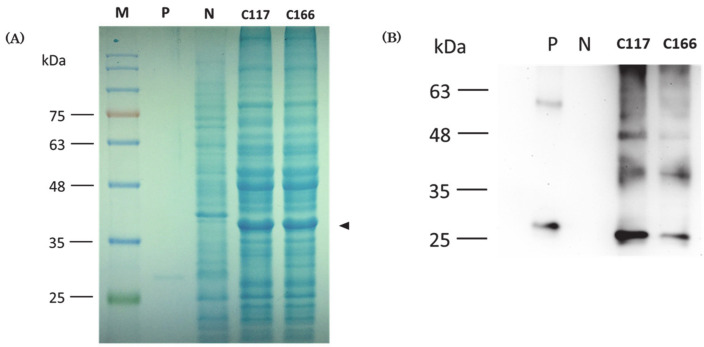
Western blot analysis of recombinant CiMAM protein expressed in transgenic *B. subtilis*. (**A**) SDS-polyacrylamide gel electrophoresis and Coomassie blue staining were used to perform the total proteins. GFP protein was the positive control (P), *B. subtilis* WB800 strain was the negative control (N), and transgenic *B. subtilis* C117 and C166 were extracted and analyzed. The recombinant protein CiMAM-F-CiMAM-GFP with molecular weight of 37 kDa produced by both transgenic *B. subtilis* strains was detected, as indicated by arrowhead. (**B**) Western blot analysis was performed to detect the recombinant protein CiMAM-F-CiMAM-GFP using polyclonal antibody against GFP. GFP protein served as the positive control (P), total proteins extracted from *B. subtilis* WB800 strain served as the negative control (N), and the total proteins from *B. subtilis* transgenic C117 and C166 strains were examined. A 37 kDa recombinant protein produced by transgenic strains was detected, as indicated by arrow.

**Figure 4 marinedrugs-19-00111-f004:**
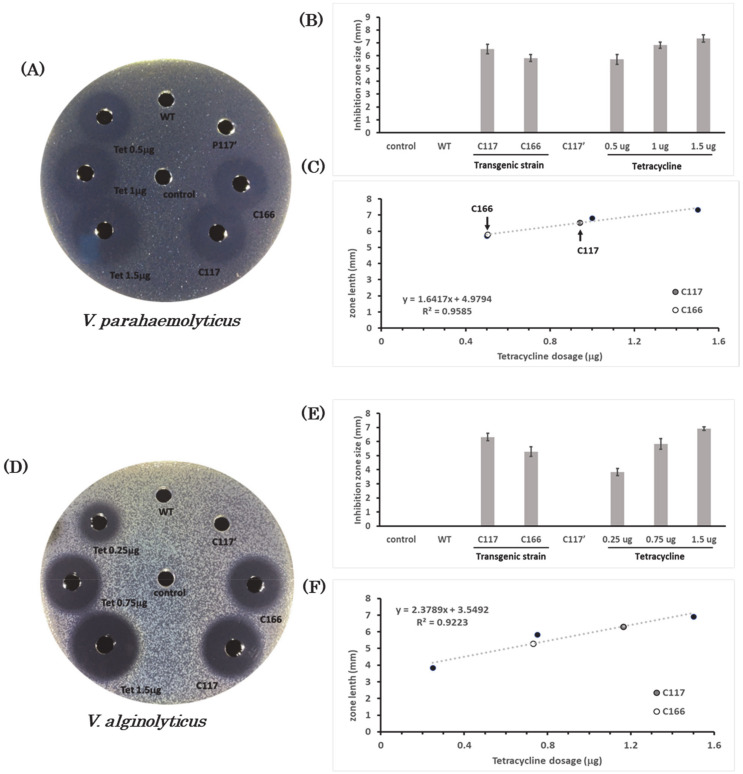
Agar well diffusion assay to demonstrate that the extract from transgenic *B. subtilis* strains exhibits bactericidal activity against halophilic pathogens. (**A**) *V. parahaemolyticus* was cultured in tryptic soy agar medium containing 2.5% NaCl and was grown as a bacterial lawn for testing survival. Tetracycline (Tet) served as a positive control, and three dosages, as indicated, were used to generate a standard curve for calculating the antibacterial potency of the extracts from *B. subtilis.* The distilled water (control) and the extract from non-transformed *B. subtilis* WB800 (WT) served as negative controls. The extract obtained from transgenic *B. subtilis* C117 without pepsin treatment (C117’) served as a mock control, while the extract from C117 treated with pepsin (C117) was an experimental group. (**B**) Each inhibition zone shown around the loading wells was measured and presented as a bar chart (*n* = 6). (**C**) Three dosages indicated on the plate were used to generate a linear trendline for obtaining a linear regression equation. Based on this equation, the equivalent amount of antibiotic potency against *V. parahaemolyticus* from extracts of transgenic *B. subtilis* C117 and C166 strains, were obtained. A similar strategy was used to examine (**D**–**F**) *V. alginolyticus* and (**G**–**I**) *V. natriegens*. (**J**,**K**) Comparison of bactericidal efficacy against *V. natriegens* between *B. subtilis* transgenic strains C117 and T13 on a salt agar plate. Four dosages of Tetracycline (Tet) with concentrations, as indicated, were used as positive controls. The extracts from T13 and C117 strains treated with pepsin were experimental groups. The data were an averaged value and presented as mean ± S.D (*n* = 6). Statistical analysis used Student’s *t*-test (**** *p* < 0.0001).

**Table 1 marinedrugs-19-00111-t001:** Linear Regression Equation of bactericidal activities of extracts from transgenic *B. subtilis* strains against pathogens.

Pathogens	Transgenic Strains: C117/C166
	LRE
*V. parahaemolyticus*	*y* = 1.6417*x* + 4.9794, *R*^2^ = 0.9585
*V. alginolyticus*	*y* = 2.3789*x* + 3.5492, *R*^2^ = 0.9223
*V. natriegens*	*y* = 1.555*x* + 5.0472, *R*^2^ = 0.9862
*E. tarda*	*y* = 2.2293*x* + 1.8928, *R*^2^ = 0.9588
*S. iniae*	*y* = 1.71*x* + 3.9767, *R*^2^ = 0.994
*S. epidermis*	*y* = 1.5157*x* + 0.7757, *R*^2^ = 0.9952

LRE: Linear Regression Equation; *R*^2^: Coefficient of determination; Extracts: treated with pepsin.

**Table 2 marinedrugs-19-00111-t002:** The bactericidal activities of extracts from transgenic *B. subtilis* strains equivalent to the efficacious doses of antibiotic against different pathogens.

**Pathogens**	**Strains**			
	C117 ^a^	C166 ^a^	T1 ^b^	T13 ^b^
	**Promoter**			
	P43	P43	P43	P43
	**Tandem repeats** ^c^			
	2× CiMAM	2× CiMAM	3× LFB	3× LFB
	**Plasmid copy number** ^d^			
	1057	349	931	647
*S. epidermidis*	48 (Amp)	41 (Amp)	154 (Amp)	130 (Amp)
*E. tarda*	39 (Amp)	37 (Amp)	79 (Amp)	33 (Amp)
*V. parahaemolyticus*	47 (Tet)	25 (Tet)	30 (Tet)	26 (Tet)
*V. natriegens*	57 (Tet)	NA	NA	9 (Tet)

Unit: ng·L^−1^, ^a^ C117 and C166: the transgenic *B. subtilis* strains harbored pP43-2CiMAM-GFP expression vector used in this study; ^b^ T1 and T13: the transgenic *B. subtilis* strain harbored pP43-6LFB-GFP expression vector published by Lee et al., 2019; ^c^ tandem repeats of cDNA encoding CiMAM or LFB (2×: double-repeats; 3×: triple repeats) in the expression vector transmitted stably in the transgenic lines; ^d^ An average copy number of plasmids per cell; (Amp) expressed as an equivalent antibacterial potency against Ampicillin; (Tet) expressed as an equivalent antibacterial potency against Tetracycline.

## Data Availability

Not applicable.

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
