# Peer review of "Using Bacillus subtilis as a Host Cell to Express an Antimicrobial Peptide from the Marine Chordate Ciona intestinalis"

_marinedrugs, 2021, doi:10.3390/md19020111_

Round 1

Reviewer 1 Report

Reviewing the article without a file with that tracks the changes is hard, but the article defintely improved.

I still think that quality of figures could be better: see for example Fig. 4 B and E: the letters are partially behind the graph

Author Response

Ms. Ref. No.:marinedrugs-1116833

Title:Using Bacillus subtilis as a host cell to express an antimicrobial peptide from the marine chordate Ciona intestinalis

Authors:Lee et al.

Reviewers' comments:

Reviewer #1

I still think that quality of figures could be better: see for example Fig. 4 B and E: the letters are partially behind the graph

Authors’ response to Q1:

Thank you. We improved them.  

Reviewer 2 Report

Dear Author,

you corrected all the issues I had underlined. Thank you very much for this work.

I have still some concern about the attempts of dosage and linear regressions used in this study in order to compare the pepsin-extracts to some purified antibiotics. Regarding the "+b" in the equation y= ax +b => Biologically it means that in absence of antibiotic you still measure a halo... this is weird isn't it ? 

Their are still some misprints or sentences hard to catch for me (sorry) :

L13 : Lines => bacterial strains
L16 : Nethertheless => suppr
L21 : CIMAM containing strains => pepsin-extracts ? 
L37 -L41 : Such -> freshwater => suppr
L51 against microbe => suppr

At this point it seems that it is a good work. It is an interesting molecular work aiming to develop a bacterial strain to be use in aquaculture farming. 

have a nice day

Author Response

Ms. Ref. No.:marinedrugs-1116833

Title:Using Bacillus subtilis as a host cell to express an antimicrobial peptide from the marine chordate Ciona intestinalis

Authors:Lee et al.

Reviewers' comments:

Reviewer #2

Dear Author,

you corrected all the issues I had underlined. Thank you very much for this work.

  1. I have still some concern about the attempts of dosage and linear regressions used in this study in order to compare the pepsin-extracts to some purified antibiotics. Regarding the "+b" in the equation y= ax +b => Biologically it means that in absence of antibiotic you still measure a halo... this is weird isn't it?

Authors’ response to Q1:

In response to your question regarding the linear regressions used in this study, we didn’t think that in absence of antibiotic one still can measure a halo if the "+b" in the equation of y= ax +b. In case in the absence of antibiotic, x=0 => y=+b, indicating that an intercept does exist. However, this theoretical point on y axis should be far away beyond the linear range obtained from the sampling points we studied. Therefore, this value is totally unreliable to count as the effective bactericidal activity. As we mentioned in the last rebuttal letter, the linear regression equations shown in this study were obtained based on the data calculated by Microsoft Excel software. Actually, this methodology was also used by other studies. For example, Chang et al. (Mentals, 2020, 10(7), 858) studied the effect of Ti-5Al-2.5Cu alloy composition on antibacterial ability and cytotoxicity at different temperatures, in which a linear regression equation y=3.766x + 27.423 (R2=0.894) was shown on Figure 9. This method was as same as what we described in this manuscript.

Their are still some misprints or sentences hard to catch for me (sorry) :

  1. L13 : Lines => bacterial strains

Authors’ response to Q2:

In response to your suggestion, we revised as follow: “Transgenic strains C117 and C166 ……..” (Please see lines 13)

  1. L16 : Nethertheless => suppr

Authors’ response to Q3:

Thank you. We deleted it.

  1. L21 : CIMAM containing strains => pepsin-extracts ? 

Authors’ response to Q4:

In response to your suggestion, we revised as follows: “….. indicating higher bactericidal activity of pepsin-extracts from rCiMAM-containing strains against halophilic bacteria compared to that fromlactoferricin-containing strains. ” (Please see lines 21-22)

  1. L37 -L41 : Such -> freshwater => suppr

Authors’ response to Q5:

Thank you. We deleted it.

  1. L51 against microbe => suppr

Authors’ response to Q6:

Thank you. We deleted it.

This manuscript is a resubmission of an earlier submission. The following is a list of the peer review reports and author responses from that submission.

Round 1

Reviewer 1 Report

The article was resubmitted to the journal without taking into big considerantions the suggestions given in the first round.

Some of the points have been improved, but many problems are still remianing:

  • results contains text that should go into material and methods
  • results are presented in a very repetitive way, and it is too easy to loose attention Even the text is repeated in the same way: compare for example lines from 223 to 228, from 240- to 244 and 257 and 262!or also 294 to 297 and 307 to 312, and again 341-344 and 357-369! I suggest to explain the procedure ones and then summarize data in a table
  • graphs quality from  figure 4 to 7 is not high, again preparing a table could help, also figures are very repetitive and do not attract attention
  • some data are presented in discussion (as it was in previous version). Why are figures 7 and 8 only discussed here? Why is the effect of salinity being discussed, while no data are present in the results section?
  • why is the concentration of TET changing? sometimes is 0.5-1 and 1.5, others is 0.25, 0.75 and 1.5. 
  • E.coli k12 is not a pathogen, while O157:H7 is

Reviewer 2 Report

Dear Colleagues,

In this work you performed the heterologous expression of a dimer of an halophilic AMP. The dimer was designed to be produce in Bacillus subtilis without harming the strains. Since the chosen AMP (CIMAM) is known to inhibit the growth of Bacilli, then, a cleavage linker (pepsin digestion) was added between the two AMP coding sequences.

The idea is to activate the molecule after its synthesis by Bacillus subtilis by a pepsin digestion. The recombinant strain could be further used in aquafarming to deal with bacterial development. Pepsin digestion within the gut of the harvested marine organisms would “activate” the treatment.

You present the following results:

  • Plasmid construction
  • Plasmid transfer and copy number assessment
  • Characterization of the AMP bioproduced
  • Antimicrobial tests

All along the text the CIMAM activity is compared to those of lactoferricin (another ribosomal produced AMP) or common antibiotics (Non ribosomal peptides).

The main results are the following:

You successfully introduced the plasmid construction in two strains of Bacillus subtilis. These latter harbored 1057 and 349 copy per cell. The fused protein CIMAM-F-CIMAM-GFP was observed on SDS-PAGE and detected through WESTERN blot analysis.

You were also successful by triggering the antimicrobial activity of B. subtilis supernatant by pepsin digestion.

The growth of numerous pathogens from freshwater to marine environments were inhibited by the pepsin treated supernatant.

Recommendation:

This work is a bit hard to read even though the approach is perfectly classic in fact. In order to gain in readiness, it should be shortened, and a more logical oriented description should be followed.

I have noticed few misprints:

Abstract

L11 : reverse the sentence ?

L14 : “strongly” seems inappropriate. In the abstract we must be more specific

Introduction:

L37-38 : shortened it is basic microbiology

L41 diffuse instead of enter?

L41-42 “synthetic antibiotic” is a notion that I don’t understand ?

L51-52 words are missing ?

L53 broad instead of board ?

L59 lactoferrincin or lactoferricin ?

Text:

In all this work numerous linear regressions are proposed. Most of them followed the pattern y=ax+b. I am not sure that “+b” is always good. It seems to me that most of the time “+b” should be equal to 0. Whatever, you should widely rework on the part dedicated the antimicrobial test and “dosages”. In some case the results observed for the supernatants are outside of the range of calibration. Weird white halos can be seen on figure 5? This part is too repetitive and long. It should be shortened and presented in a table. It could be illustrated with few representative plates. The whole approach could be shown in supplementary data.

L458 Table 1 is unclear for me?

L527 figure 8 could be shown in supplementary data

At this stage the work is not presented well enough to be fully understood and evaluated. I would recommend to shortened and clarify the preceding issues  before being published.

good luck